# Screening and Optimization of Conditions for the Adsorption of Cd^2+^ in Serpentine by Using Response Surface Methodology

**DOI:** 10.3390/ijerph192416848

**Published:** 2022-12-15

**Authors:** Xufeng Zhang, Liyu Du, Wenjuan Jin

**Affiliations:** College of Land and Environment, Shenyang Agricultural University, Shenyang110161, China

**Keywords:** serpentine, Cd^2+^, adsorption, response surface methodology (RSM)

## Abstract

In order to explore the optimal conditions for the adsorption of Cd^2+^ in serpentine, this paper studied the adsorption of simulated cadmium solutions with serpentine as an adsorbent. On the basis of a single factor experiment, four factors including the amount of serpentine, initial pH, the initial concentration of solutions, and adsorption time were selected as the influencing factors, and the adsorption quantity and adsorption rate of serpentine to Cd^2+^ were double response values using the Box–Behnken design. Response surface analyses were used to study the effects of four factors on the adsorption quantity and adsorption rate of serpentine on cadmium, and the interaction between various factors. The results showed that the optimum adsorption conditions were as follows: the amount of serpentine was 1%, the initial pH was 5.5, the initial solution concentration was 40.83 mg·L^−1^, and the adsorption time was 26.78 h. Under these conditions, the theoretical adsorption quantity and adsorption rate of serpentine to Cd^2+^ were 3.99 mg·g^−1^ and 95.24%, respectively. At the same time, after three repeated experiments, the actual adsorption quantity and adsorption rate of serpentine to Cd^2+^ were 3.91 mg·g^−1^ and 94.68%, respectively, and the theoretical value was similar to the actual value. Therefore, it was proved that the experimental design of the regression model is reliable, and it is feasible to use the response surface method to optimize the adsorption conditions of serpentine on Cd^2+^.

## 1. Introduction

Industrial wastewater is an important source of pollution relative to the water environment. With the acceleration of industrialization processes, a large amount of industrial heavy metal wastewater containing toxic chemical elements, such as cadmium (Cd), zinc (Zn), and nickel (Ni), enters the environment [1,2]. Among the known heavy metals, Cd is one of the most toxic due to its bioaccumulation in human and animal bodies; moreover, it is almost non-biodegradable, even at low concentrations [3,4]. The majority of Cd contamination in water and wastewater originates from ore processing plants, metal refineries’ discharges, mine drainage water, waste batteries’ runoff, the manufacture of phosphate fertilizers and pesticides, pigment production, printing, the photographic industry, and rainwater runoff from mining areas [5,6]. In aqueous media, Cd is shown as Cd^2+^, a highly mobile divalent heavy metal ion that can be easily carried by water and wastewater, leading to the widespread release of this poisonous element [7]. Therefore, methods for effectively treating heavy metal industrial wastewater are urgently needed.

The treatment methods of heavy metal wastewater generally include adsorption, chemical precipitation, redox, ion exchange, membrane separation, electrochemical methods, etc. [8]. Among them, the adsorption method uses porous solid adsorbents to treat wastewater [9], which has the following advantages: low cost, good treatment effects, and no secondary pollution [10,11]. Moreover, it is currently one of the heavy metal wastewater treatment methods that has been popularized and possesses application value. The key to the adsorption method is the choice of adsorbents, and there are various types of adsorbents. Among these adsorbents, silicate minerals have the advantages of abundant reserves, ample sources, and low cost, and they have a large specific surface area and a special crystal layer structure [12,13]. Silicate minerals can adsorb heavy metals, mainly including surface adsorption and ion exchange. The silicate mineral itself is an excellent adsorbent that has a large specific surface area and surface energy, which can adsorb metal ions on its surface. Ion exchanges refer to the use of interlayer metal examples in silicate minerals, such as Na^+^, Al^3+^, and other metal ions for ion exchange with heavy metal ions; this process belongs to electrostatic adsorption [14,15]. Therefore, it has great potential in the treatment of heavy metal wastewater treatment. Silicate minerals mainly include serpentine, sepiolite, zeolite, bentonite, etc. 

Serpentine has a large specific surface area, good pore structure, and strong surface activity [16]. It is a hydrous magnesium-rich silicate mineral, containing 43% MgO, 44.1% SiO_2_, 12.9% water, and, usually, some small amounts of other elements [17]. Nevertheless, the stabilization performance of the original serpentine on heavy metals is limited due to finite binding sites, thus hindering its feasibility in application [18]. The modification methods of materials vary, including methods such as doping foreign substances, thermal modification, etc. For example, some studies have reported that the introduction of N doping in carbon materials can alter the electron density of partial C atoms, and the specific surface area of carbon materials can be improved, thereby improving their adsorption and oxidation capacity for pollutants [19]. The characteristics of serpentine adsorbing heavy metals after thermal modification have been studied, and it has been confirmed that thermal activation treatments can increase the specific surface area of serpentine and reorganize its internal structure, greatly improving its ability to adsorb heavy metal ions [20,21]. For this reason, considering the thermal activation modification of serpentine, the purpose is to improve its ability to adsorb heavy metals. Therefore, methods for optimizing and screening the modified serpentine and the adsorption conditions in order to achieve the best repair effect are important for the further development of serpentine as a good repair agent. 

Response surface methodology (RSM) is a method oriented toward experimental designs and mathematical modeling optimizations [22]. According to a small number of representative experiments, a mathematical model of various factors and experimental results was established to obtain the best prediction figure of merit and the optimization of process conditions, and the Box–Behnken central combination design was a commonly used response surface method for finding optimal conditions in a multi-factor system [23,24]. Currently, it has been widely used in many fields [25,26], and there have been relatively few studies on the application of response surface methods for optimizing the process of absorbing heavy metals in wastewater by serpentine. 

Therefore, from the perspectives of serpentine structure and surface property changes, this article explores the adsorption capacity of Cd^2+^ under different conditions using static adsorption experiments. On the basis of the single factor experiment, the Box–Behnken experimental design is used to optimize the conditions for the adsorption of Cd^2+^ by serpentine in order to provide a theoretical basis for improving the removal effect of Cd^2+^ and to provide an optimized process for the treatment of Cd^2+^ in wastewater.

## 2. Materials and Methods

### 2.1. Test Materials

Natural serpentine (S-0): The serpentine was obtained from Xiuyan County, Anshan City, Liaoning Province, China; ground by hand; passed through a 200-mesh sieve; and stored in a plastic sealed bag for later use. The main compositions and contents of serpentine are shown in the Table 1. 

Thermally modified serpentine (S-700): The natural serpentine was placed in a muffle furnace and calcined at a constant temperature of 700 °C for 2 h at a heating rate of 5 °C/min. All serpentine used in this article comprised thermally modified serpentine.

The superficial morphologies of materials were identified by using a scanning electron microscope (SEM). An energy dispersive spectrometer (EDS) was used to determine the element content of the sample, and the crystal textures of the two samples were detected via an X-ray powder diffractometer (XRD) [27].

### 2.2. Experimental Method

#### 2.2.1. Equilibrium Adsorption Experiment

In this study, 0.5 g of natural serpentine (S-0) and thermally modified serpentine (S-700) was dispensed into a 50 mL centrifuge tube, and then 50 mL of cadmium solution was added at different concentrations. The initial pH of the solution was 6, and the concentrations of the cadmium solution were 40 mg·L^−1^, 80 mg·L^−1^, 120 mg·L^−1^, 160 mg·L^−1^, and 200 mg·L^−1^. These centrifugal tubes were placed in a high-speed constant-temperature oscillator for oscillation, with a rotating speed of 220 r·min^−1^. The temperatures of 25 °C, 35 °C, and 45 °C were used, and an oscillation duration of 24 h was used. Centrifugation and filtration occurred after the oscillations.

#### 2.2.2. Single-Factor Experiment

After the pre-experiment, the amount of serpentine (AS), the initial pH (pH), the initial concentration of the solution (C), and the adsorption time (T) were selected to explore the influence of Cd^2+^ adsorption quantity (AQ) and adsorption rate (AR), and the best range for each influencing factor was preliminarily determined.

The pH was 6, C was 50 mg·L^−1^, and T was 24 h. The influence of different AS (1%, 2%, 3%, 4%, and 5%) on the AQ and AR was investigated.

The AS was 1%, C was 50 mg·L^−1^, and T was 24 h. The influence of different pH (2, 3, 4, 5, and 6) on the AQ and AR was investigated.

The AS was 1%, the pH was 6, and T was 24 h. The influence of different C (10 mg·L^−1^, 20 mg·L^−1^, 50 mg·L^−1^, 100 mg·L^−1^, and 200 mg·L^−1^) on the AQ and AR was investigated.

The AS was 1%, the pH was 6, and C was 50 mg·L^−1^. The influence of different T (0.5 h, 2 h, 6 h, 12 h, and 24 h) on the AQ and AR was investigated.

Each of the above single-factor experiments was carried out three times.

#### 2.2.3. Box–Behnken Center Combination Experiment Design

Based on the single factor experiment, a Box–Behnken design of the central combination experiment was adopted, and four factors of AS, pH, C, and T were selected. The AQ and AR were obtained relative to the Cd^2+^ double response value, and a response surface analysis experiment was designed with four factors and three levels. The three levels of each factor were coded with −1, 0, and 1, as shown in Table 2.

### 2.3. Data Analysis

The concentration of Cd^2+^ in the solution was measured by an atomic absorption spectrophotometer (WYA2200); then, the adsorption quantity and adsorption rate were calculated by using the following formula.

Adsorption quantity:(1)$$\;AQ=\frac{v\left(Co-Ce\right)}{m}$$

Adsorption rate:(2)$$AR=\frac{Co-Ce}{Co}\times 100\%$$

Here, in Equations (1) and (2), v denotes the volume of the added Cd^2+^ solution, L; *m* denotes the amount of the added serpentine, g; *Co* and *Ce* denote the Cd^2+^ concentration in the solution before and after adsorption, mg·L^−1^; *AQ* denotes the adsorption quantity, mg·g^−1^; *AR* denotes the adsorption rate, %.

The experimental data were analyzed and processed by Origin8.5 and Design Expert12.0.

## 3. Results and Discussion

### 3.1. Serpentine Sample Characterization

Natural serpentine (S-0) comprised many irregular flaky structures, while the thermally modified serpentine (S-700) had no layered structure, and the surface of the material comprised a block structure formed by some agglomerated particles (Figure 1).

After the thermal modification of serpentine, the content of oxygen (O) decreased, while the content of magnesium (Mg) and silicon (Si) increased, which indicated that the serpentine was dehydrogenated during the thermal activation process (Figure 2).

For natural serpentine (S-0), the diffraction peaks (JCPDS 07-0417) attributed to antigorite (a type of serpentine) appeared at 2θ of 12.1°, 24.5°, 35.6°, 37.2°, 41.6°, 42.0°, and 50.2°. When the modification temperature was 700 °C, compared with natural serpentine (S-0), its diffraction peaks obviously changed, and all characteristic diffraction peaks attributed to antigorite disappeared (Figure 3). At the same time, some new diffraction peaks appeared at 2θ of 22.9°, 32.3°, 35.5°, 35.7°, 39.7°, and 52.3°. These diffraction peaks belonged to the characteristic diffraction peaks of forsterite (JCPDS 34-0189), indicating that the crystal structure of serpentine changed when the modification temperature was 700 °C; that is, the serpentine was transformed into forsterite.

Compared with natural serpentine (S-0), the specific surface area and pore volume of thermally modified serpentine (S-700) significantly improved, which was mainly due to the collapse of the layered structure of serpentine at high temperatures and the formation of accumulated slit pores (Table 3).

### 3.2. Study on the Adsorption Isotherms of Cd^2+^ on Serpentine

A Langmuir isothermal adsorption model was used to fit the adsorption process of natural serpentine (S-0) and heat-modified serpentine (S-700) for Cd^2+^ in order to explore the adsorption characteristics of serpentine for Cd^2+^.

Langmuir isothermal adsorption model: $$\frac{C_{e}}{q_{e}}=\frac{C_{e}}{q_{m}}+\frac{1}{k_{L}q_{m}}$$

*C_e_* denotes the balance concentration, mg·L^−1^; *q_e_* denotes the balance adsorption quantity, mg·g^−1^; *q_m_* denotes the maximum adsorption quantity, mg·g^−1^; *K_L_* denotes the adsorption model constant.

The Langmuir isotherm adsorption model can describe the entire adsorption process, indicating that serpentine and Cd^2+^ experienced monolayer adsorption (Figure 4, Table 4). The maximum adsorption quantity of thermally modified serpentine (S-700) was higher than that of natural serpentine (S-0), indicating that the adsorption effect of thermally modified serpentine (S-700) on Cd^2+^ significantly increased. In addition, when the temperature increased from 25 °C to 45 °C, the adsorption quantity of serpentine to Cd^2+^ increased, indicating that increasing the temperature within a certain temperature range promoted the adsorption.

### 3.3. Single Factor Experiment Results

#### 3.3.1. Effect of the AS on Adsorption Effect

The effect of different ASs on the AQ and AR of Cd^2+^ is shown in Figure 5. It can be observed that the AQ of Cd^2+^ gradually decreased as the AS increased. Specifically, when the AS was 1%, its AQ of Cd^2+^ reached 4.13 mg·g^−1^, but when the AS was 2%, the AQ of Cd^2+^ reduced to 2.48 mg·g^−1^, which was 40% lower than that of 1%. When the AS was 5%, the AQ of Cd^2+^ was only 0.99 mg·g^−1^, which was 76% lower than that of 1%. This change in AQ may be due to the increase in the AS, although the adsorption sites provided by serpentine gradually increased, but the AQ of Cd^2+^ that can be adsorbed per unit area would decrease, resulting in a decrease in AQ. This was similar to the research results of Sun Pengcheng, who used serpentine to adsorb Cd^2+^ in water [28]. Although the AQ of Cd^2+^ in water gradually decreased with the increase in the AS, the AR of Cd^2+^ increased. When the AS was 1%, although the AQ of Cd^2+^ was the largest, the AR of Cd^2+^ was low at this moment, which was only 82.66%. When the AS was 2%, the AR of Cd^2+^ had risen to 99.28%. At this time, the AR reached a very high level. When the AS increased to 3%, 4%, and 5%, it was found that the increase in the AR was not significant and it was maintained at a high level. Under this condition, the AQ of Cd^2+^ also reached a high level, which was 2.48 mg·g^−1^. In summary, the AS was selected as 2%.

#### 3.3.2. Effect of the pH on Adsorption Effect

The effect of different pH on the AQ and AR of Cd^2+^ is shown in Figure 6. It can be seen that the AQ and AR of Cd^2+^ gradually increased when the pH increased. Specifically, when the pH was 2, the AQ of Cd^2+^ was 2.96 mg·g^−1^, and when the pH increased to 6, the AQ also increased to 4.10 mg·g^−1^, which comprised an increase of 28% compared to when the pH was 2. When the pH was 2, the AR of Cd^2+^ was 59.13%. When the pH increased to 6, the AR of Cd^2+^ also increased to 81.93%. Compared with when the pH was 2, it increased by 38%. It could be seen that when the pH increased from 2 to 6, the AQ and AR of Cd^2+^ greatly improved, and the change in pH had a significant effect on adsorption processes. This change in AQ and AR might be because of the phenomenon where, with an increase in the pH, deprotonation occurred on the surface of the serpentine, and the number of negatively charged sites increased, resulting in an increase in the adsorption of Cd^2+^ on the serpentine [29]. Studies have also shown that adsorbents showed different surface charges in different pH solutions, which led to different removal effects for the heavy metals. For example, studies have shown that the charges on biochemical composite (CFB1-P) surfaces were negative at a pH of over 2.11, while they were positive at a pH beneath 2.11 [30]. Subsequently, measuring the zero potential of serpentine and analyzing the results should be considered. When the pH was 6, the AQ (4.10 mg·g^−1^) and AR (81.93%) of Cd^2+^ by serpentine reached the highest values, so the pH was selected at 6 for subsequent experiments. 

#### 3.3.3. Effect of the C on Adsorption Effect

The effect of different Cs on the AQ and AR of Cd^2+^ is shown in Figure 7. It can be observed that the AQ of Cd^2+^ continuously increased with the increase in C. Specifically, when C was 10 mg·L^−1^, the AQ of Cd^2+^ was only 0.99 mg·g^−1^, but when the C was 50 mg·L^−1^, the AQ of Cd^2+^ increased to 4.15 mg·g^−1^. Compared with the C at 10 mg·L^−1^, it increased by 3.2 times. At this time, it can be observed that there was a significant increase. When the C continued to increase to 200 mg·L^−1^, it could be seen that the AQ of Cd^2+^ increased to 9.53 mg·g^−1^, compared with the C at 10 mg·L^−1^ that increased by 8.6 times; the increase was extremely significant. Although the AQ of Cd^2+^ increased with the increase in C, the AR of Cd^2+^ exhibited opposite results. It could be seen that when the C was 10 mg·L^−1^, the AQ was the smallest, but at this time, the AR reached 99.74%, but, when the C increased to 50 mg·L^−1^, the AR of Cd^2+^ was 82.91%, which was 17% lower than when the C was 10 mg·L^−1^. When the C reached 200 mg·L^−1^, the AQ of Cd^2+^ was the largest, but at this time, the AR was only 47.63%; this may be explained by the continuous increase in C, which would cause the AR of Cd^2+^ to continually decline. This change in the AR of Cd^2+^ in water might be due to the fact that the adsorption sites on the surface of serpentine were certain. When the C was low, the adsorption equilibrium could provide adsorption sites that were relatively sufficient, and the adsorption rate was high; when the C was high, the adsorption sites gradually became saturated, resulting in a gradual decrease in the AR [29]. The coexistence of multiple ions might affect the adsorption effect of serpentine, and, in order to better explore the potential of serpentine in treating cadmium-containing industrial wastewater, the coexisting ion test will be added in subsequent experiments [31]. Considering the change law and value of both AQ and AR, when the C was 50 mg·L^−1^, the AQ and AR of Cd^2+^ in water were 4.15 mg·g^−1^ and 82.91%, respectively; they were relatively high. Therefore, the C selected for subsequent experiments was 50 mg·L^−1^.

#### 3.3.4. Effect of the T on Adsorption Effect

The effect of different T on the AQ and AR of Cd^2+^ is shown in Figure 8. It can be observed that the AQ and AR of Cd^2+^ both decreased first and then increased with the increase in the T. When the T was 0.5 h, the AQ of Cd^2+^ was 3.17 mg·g^−1^; when the T reached 2 h, the AQ of Cd^2+^ was 2.91 mg·g^−1^. Compared with the adsorption time of 0.5 h, it was reduced by 8%. When the T reached 24 h, the AR of Cd^2+^ was 4.10 mg·g^−1^, which increased by 41% compared with when the T was 2 h. When the T was 0.5 h, the AR of Cd^2+^ was 63.40%. When the T was 2 h, the AR of Cd^2+^ was 58.11%, which was 8% lower than when the T was 0.5 h. When the T was 24 h, the AR of Cd^2+^ was 81.93%, which was increased by 41% when the adsorption time was 2 h. The adsorption was rapid in the first 0.5 h, which might be because of the observation that the adsorption at this stage belonged to the nonspecific adsorption phase, and the stability efficiency was high and labile [32]. In the next few hours, the combination of the driving force of metal ions and the potential weak retention ability of serpentine resulted in the re-release of metal ions to a certain extent, thereby reducing the AQ and AR [33]. After the adsorption time exceeded 10 h, the AQ and AR of Cd^2+^ gradually increased. When the adsorption time was 24 h, the AQ and AR of Cd^2+^ reached the highest, respectively, at 4.10 mg·g^−1^ and 81.93%. In summary, all subsequent experiments used a T of 24 h.

### 3.4. Experiment Design and Results of RSM

#### 3.4.1. Experiment Design of RSM

Taking the AQ and AR of Cd^2+^ as response values with the AS (A), pH (B), C (C), and T (D) being independent variables, a four-factor three-level central combination design, including 29 experimental programs, was established. Each experiment was repeated three times and averaged. The specific experimental programs are shown in Table 5.

Design-Expert 12.0 was used to perform regression analyses on the experimental data in Table 5, and the quadratic polynomial regression equation was obtained as follows.
Y1 = 2.48 − 1.36A − 0.0158B + 0.4167C + 0.0833D + 0.0625AB − 0.09AC − 0.18AD − 0.005BC − 0.02BD − 0.005CD + 0.5765A^2^ + 0.009B^2^ − 0.0748C^2^ − 0.0573D^2^
Y2 = 99.28 + 6.94A − 0.1642B − 2.93C + 0.865D + 1.14AB + 3.39AC + 0.26AD −0.2025BC − 0.7675BD − 0.2025CD − 5.47A^2^ + 0.2983B^2^ − 1.05C^2^ − 3.57D^2^

The results of the analysis of variance based on the regression model of the AQ of Cd^2+^ are shown in Table 6. It could be seen that the *p*-value of the regression model was less than 0.0001, reaching an extremely significant level, indicating that the model had practical significance. The regression model resulted in R² = 0.9955 and R^2^_Adj_ = 0.9910, indicating that the model could fit the experimental data well and could be used to predict the effect of the AQ of Cd^2+^. It can also be seen from Table 6 that the *p*-values of the AS(A) and the C (C) were all less than 0.0001, reaching extremely significant levels. The *p*-values of the pH (B) and the T (D) were not significant. It can be inferred that the main and secondary factors affecting the AQ of Cd^2+^ are as follows: A > C > D > B.

The results of analysis of variance based on the regression model of the AR of Cd^2+^ are shown in Table 7. It can be observed that the *p*-value of the regression model is less than 0.0001, reaching an extremely significant level, indicating that the model had practical significance. The regression model resulted in R² = 0.9376 and R^2^_Adj_ = 0.8752, indicating that the model could fit the experimental data well and can be used to predict the effect of the AR of Cd^2+^. It can also be observed in Table 7 that the *p*-values of the AS (A), the C (C), and the T (D) were all less than 0.01, and all reached significant levels, while the pH (B) was not significant. It can be inferred that the main and influential factors of serpentine on the AR of Cd^2+^ are as follows: A > D > C > B.

#### 3.4.2. Response Surface Analysis of the Interaction Influence of Various Factors 

RSM uses reasonable experimental design to find the optimal process parameters via the analysis of the response surface contour, and it can intuitively evaluate the interaction between various factors using an analysis of the 3D surface graph [34,35,36]. In the 3D surface graph, the slope of the curved surface indicates the degree of influence of the experimental factor on the response value. The higher the slope, the more significant the influence of the experimental factor on the response value [37,38].

The effect of the interaction between the four factors on the AQ of Cd^2+^ is shown in Figure 9. From these 3D surface graphs, it can be observed that the degree of inclination of the surfaces of graphs (b) and (c) was larger than that of the remaining four graphs, indicating that the interaction between AC and AD had a significant effect on the AQ of Cd^2+^, while the interaction between other factors had no obvious effects on the AQ of Cd^2+^. 

The effect of the interaction between the four factors on the AR of Cd^2+^ is shown in Figure 10. From these 3D surface graphs, it can be seen that the degree of inclination of the graph (b) was the largest. Then, the slopes of the curved surfaces of the graphs (a) and (c) were the second largest. These were all larger than the curved surfaces of the remaining three graphs, indicating that the interaction between AC had the most obvious impact on the AR of Cd^2+^, and the interaction between AB and AD also had a significant effect on the AR of Cd^2+^, while the interaction between the other factors had no obvious effects on the AR of Cd^2+^.

### 3.5. Determination of Optimal Adsorption Conditions and Verification Experiments

Using the above analysis, a set of prediction data was finally obtained, namely the optimal adsorption conditions for serpentine to adsorb Cd^2+^. Specifically, the AS was 1%, the pH was 5.5, the C was 40.83 mg·L^−1^, and the T was 26.78 h. Under these conditions, the AQ of Cd^2+^ was 3.99 mg·g^−1^, and the AR of Cd^2+^ was 95.24%. Then, in order to verify the accuracy of the prediction data, a set of verification experiments was designed. The values of the four factors were set: The AS was 1%, the pH was 5.5, the C was 41 mg· L^−1^, and the T was 26.8 h. The results are shown in Table 6. It can be observed in Table 8 that the AQ of Cd^2+^ was 3.91 mg·g^−1^, and the AR of Cd^2+^ was 94.68%. The data obtained by the verification experiment are similar to the data predicted by the software, which show that optimizing the adsorption conditions of serpentine to Cd^2+^ by RSM is feasible.

## 4. Conclusions

Based on the single factor experiment, the Box–Behnken RSM was adopted and it was analyzed according to Design Expert 12.0. The optimal adsorption conditions obtained by optimizing the factors affecting the adsorption of Cd^2+^ by serpentine were as follows: The AS was 1%, the pH was 5.5, the C was 40.83 mg·L^−1^, and the T was 26.78 h. Under these conditions, the AQ of Cd^2+^ was 3.99 mg·g^−1^, the AR of Cd^2+^ was 95.24%, and the experiment had good repeatability. This condition could maximize the adsorption quantity of serpentine on the premise of ensuring a high adsorption rate of Cd^2+^. The acquisition of this experimental condition could provide further theoretical support for improving the purification efficiency of heavy metal wastewater and for exploring the treatment of cadmium-containing wastewater by serpentine.

## Figures and Tables

**Figure 1 ijerph-19-16848-f001:**
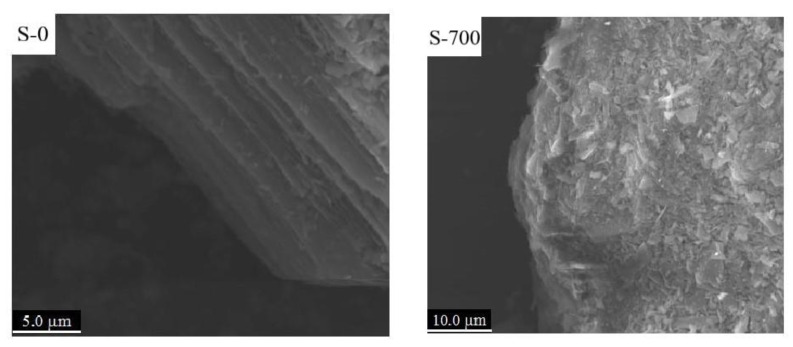
SEM images of two serpentines.

**Figure 2 ijerph-19-16848-f002:**
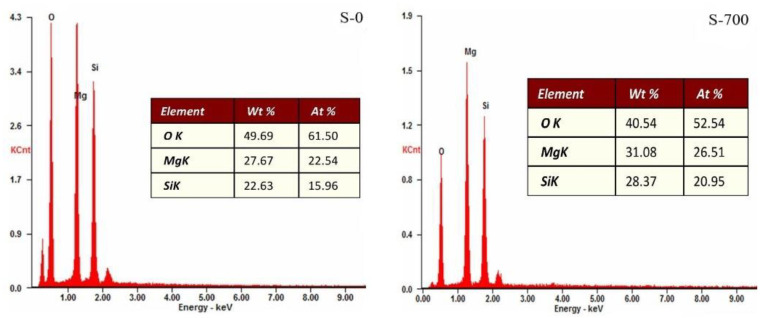
EDS images of two serpentines.

**Figure 3 ijerph-19-16848-f003:**
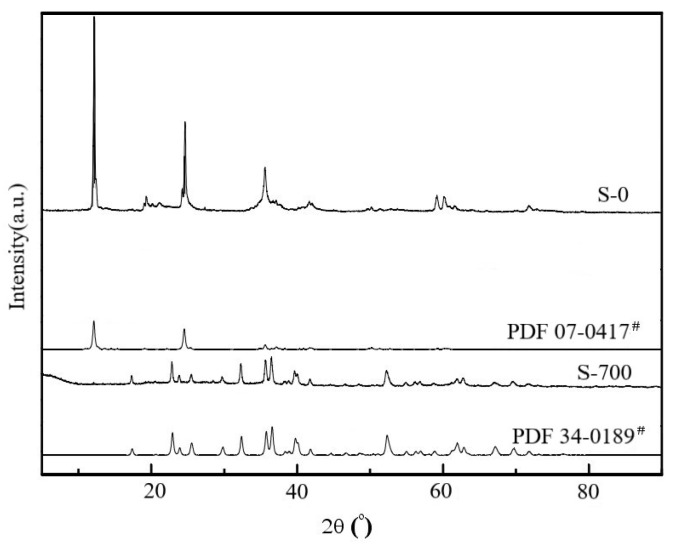
XRD patterns for two serpentines.

**Figure 4 ijerph-19-16848-f004:**
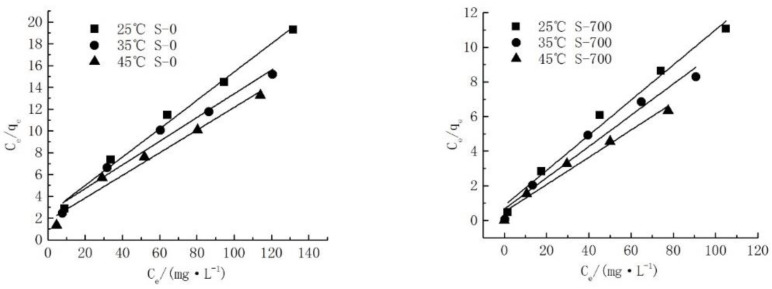
Langmuir isothermal adsorption model of serpentine to Cd^2+^.

**Figure 5 ijerph-19-16848-f005:**
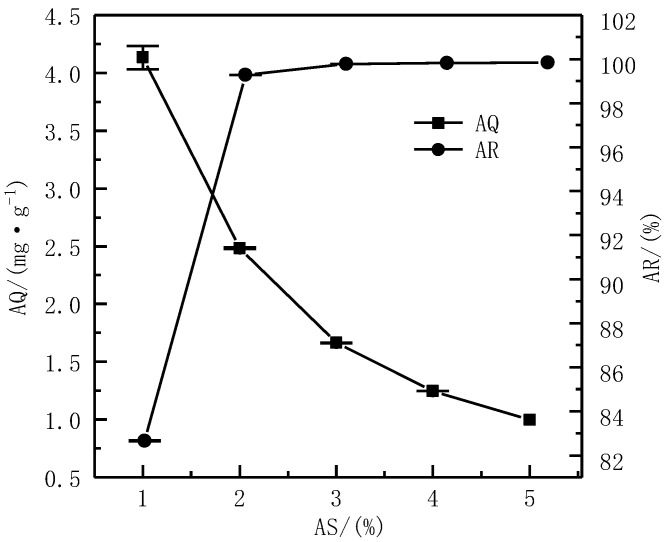
Effect of the AS on adsorption effect.

**Figure 6 ijerph-19-16848-f006:**
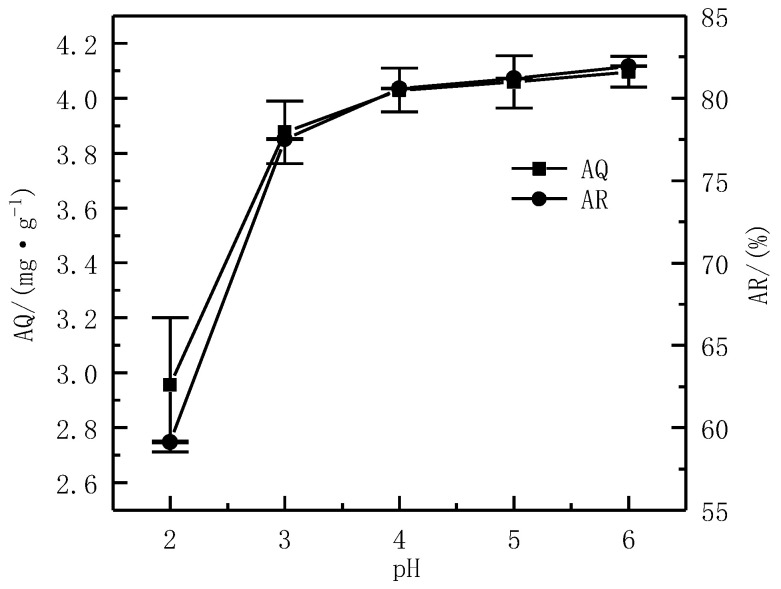
Effect of the pH on adsorption effect.

**Figure 7 ijerph-19-16848-f007:**
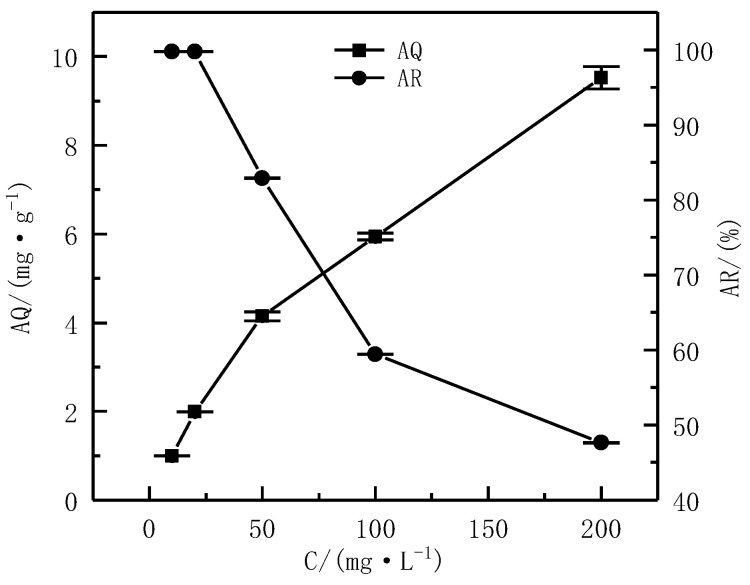
Effect of the C on adsorption effect.

**Figure 8 ijerph-19-16848-f008:**
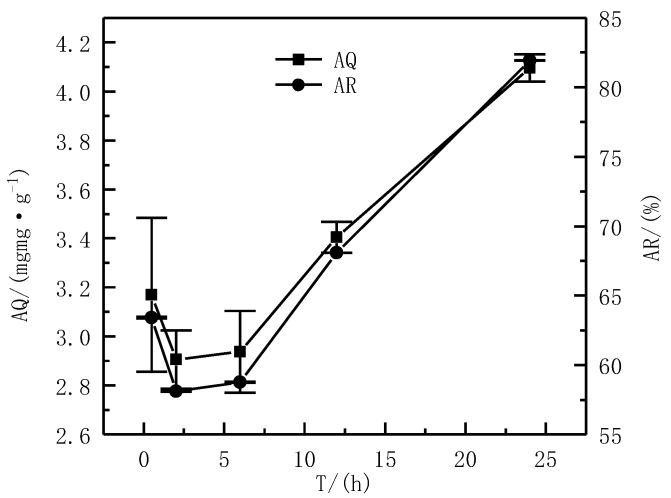
Effect of the T on adsorption effect.

**Figure 9 ijerph-19-16848-f009:**
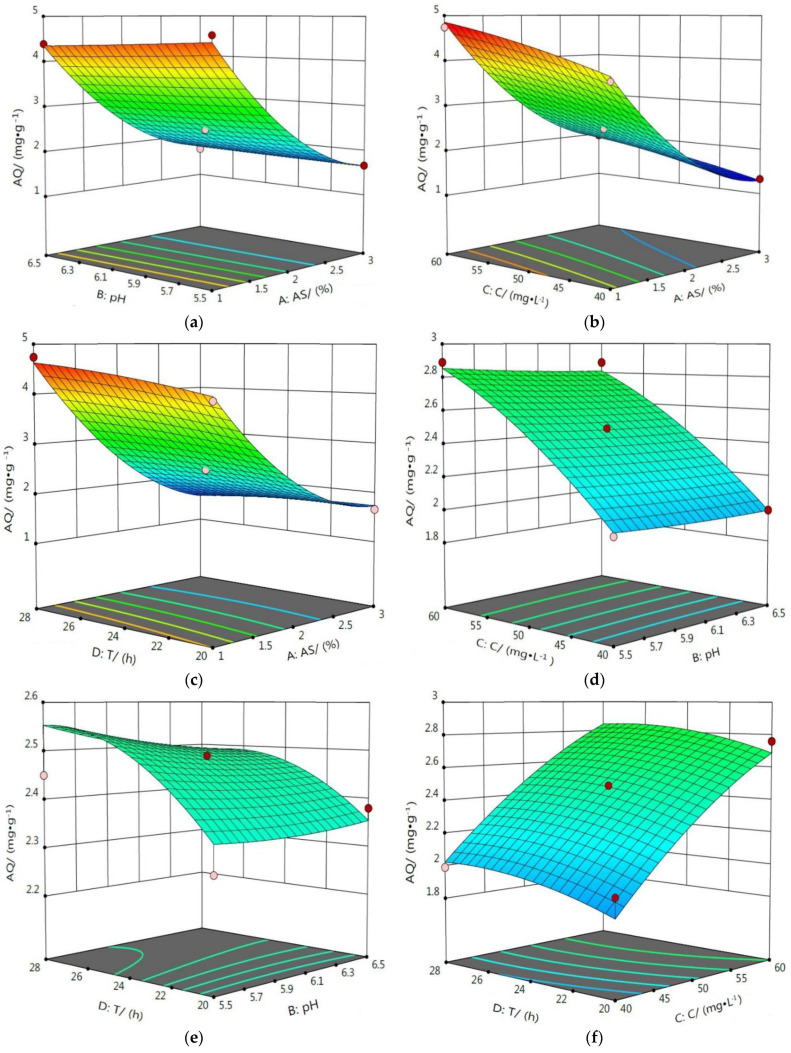
Response surface diagram of the effect of the interaction of various factors on the AQ of Cd^2+^. Response surface diagram of interaction between AS and pH (**a**), AS and C (**b**), AS and T (**c**), pH and C (**d**), pH and T (**e**), C and T (**f**) on the AQ of Cd^2+^.

**Figure 10 ijerph-19-16848-f010:**
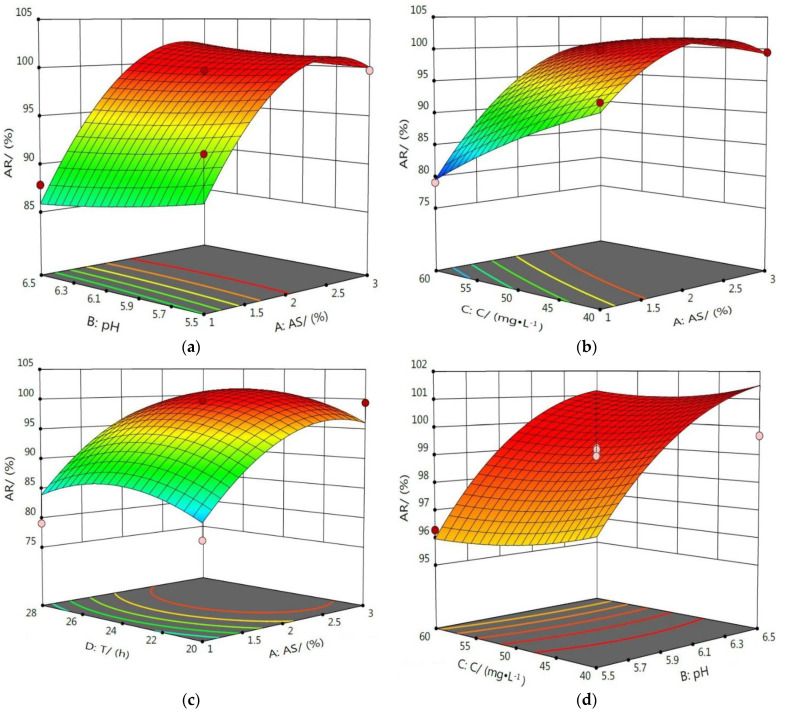
Response surface diagram of the interaction of various factors on the AR of Cd^2+^. Response surface diagram of interaction between AS and pH (**a**), AS and C (**b**), AS and T (**c**), pH and C (**d**), pH and T (**e**), C and T (**f**) on the AR of Cd^2+^.

**Table 1 ijerph-19-16848-t001:** The main composition and content of serpentine.

Composition	SiO_2_	MgO	CaO	Al_2_O_3_	Fe_2_O_3_
Content (%)	52.93	46.52	0.85	0.34	0.36

**Table 2 ijerph-19-16848-t002:** Factors and levels in response surface methodology.

Factor	Level		
	−1	0	1
A AS (%)	1	2	3
B pH	5.5	6	6.5
C C (mg·L^−1^)	40	50	60
D T (h)	20	24	28

**Table 3 ijerph-19-16848-t003:** Structural properties of serpentines.

S-T	BET Surface Area (m^2^ g^−1^)	Pore Volume (cm^3^ g^−1^)	Aperture (nm)
S-0	6.1	0.017	11.01
S-700	13.7	0.028	8.10

**Table 4 ijerph-19-16848-t004:** Fitting parameters of the Langmuir isotherm adsorption model.

S-T	Temperature	Langmuir		
		R^2^	q_m_/(mg∙g^−1^)	*K_L_*
S-0	25 °C	0.9883	7.6982	0.0536
	35 °C	0.9651	9.1971	0.0428
	45 °C	0.9670	9.6246	0.0585
S-700	25 °C	0.9816	9.8774	0.1185
	35 °C	0.9527	11.0693	0.1379
	45 °C	0.9696	12.6904	0.1594

**Table 5 ijerph-19-16848-t005:** Experiment design and results of RSM.

Experimental No.	A	B	C	D	Y1 AQ (mg·g^−1^)	Y2 AR (%)
1	−1	−1	0	0	4.65 ± 0.0803	92.99 ± 1.61
2	0	−1	−1	0	1.99 ± 0.0008	99.39 ± 0.04
3	−1	0	0	1	4.33 ± 0.0243	86.56 ± 0.49
4	0	0	0	0	2.49 ± 0.0037	99.75 ± 0.15
5	0	0	0	0	2.48 ± 0.0055	99.21 ± 0.22
6	−1	1	0	0	4.39 ± 0.0254	87.84 ± 0.51
7	0	0	1	−1	2.76 ± 0.0295	92.10 ± 0.98
8	0	0	−1	1	1.99 ± 0.0018	99.29 ± 0.09
9	0	1	0	1	2.46 ± 0.0172	98.58 ± 0.69
10	1	0	1	0	1.97 ± 0.0015	98.59 ± 0.07
11	0	1	1	0	2.87 ± 0.0033	95.78 ± 0.11
12	0	1	−1	0	1.99 ± 0.0004	99.70 ± 0.02
13	1	1	0	0	1.65 ± 0.0031	99.17 ± 0.18
14	0	−1	1	0	2.89 ± 0.0122	96.28 ± 0.41
15	1	0	0	−1	1.66 ± 0.0006	99.41 ± 0.03
16	−1	0	−1	0	3.74 ± 0.0466	93.53 ± 1.17
17	0	1	0	−1	2.38 ± 0.0152	95.23 ± 0.61
18	0	−1	0	1	2.45 ± 0.0085	98.14 ± 0.34
19	0	0	−1	−1	1.96 ± 0.0069	98.10 ± 0.35
20	1	−1	0	0	1.66 ± 0.0003	99.75 ± 0.02
21	0	−1	0	−1	2.29 ± 0.0044	91.72 ± 0.18
22	1	0	−1	0	1.33 ± 0	99.49 ± 0
23	0	0	0	0	2.48 ± 0.0121	99.14 ± 0.48
24	1	0	0	1	1.66 ± 0.0006	99.45 ± 0.04
25	−1	0	0	−1	4.02 ± 0.1325	80.46 ± 2.65
26	0	0	0	0	2.48 ± 0.0005	99.03 ± 0.02
27	0	0	1	1	2.77 ± 0.0301	92.48 ± 1.00
28	−1	0	1	0	4.74 ± 0.1982	79.06 ± 3.30
29	0	0	0	0	2.48 ± 0.0032	99.27 ± 0.13

**Table 6 ijerph-19-16848-t006:** Analysis of variance in regression model based on serpentine AQ of Cd^2+^.

Source of the Variance	Square-Sum	Degree of Freedom	Average Variance	F Value	*p* Value	Significancy
Model	25.68	14	1.83	222.24	<0.0001	**
A	21.2	1	21.2	2568.79	<0.0001	**
B	0.0026	1	0.0026	0.3153	0.5833	
C	2.10	1	2.10	254.23	<0.0001	**
D	0.0285	1	0.0285	3.46	0.0842	
AB	0.0154	1	0.0154	1.86	0.1940	
AC	0.0319	1	0.0319	3.86	0.0696	
AD	0.0232	1	0.0232	2.81	0.1160	
BC	0.0001	1	0.0001	0.0137	0.9083	
BD	0.0015	1	0.0015	0.1782	0.6793	
CD	0.0000	1	0.0000	0.0047	0.9461	
A^2^	1.91	1	1.91	230.86	<0.0001	**
B^2^	0.0050	1	0.0050	0.6097	0.4479	
C^2^	0.0212	1	0.0212	2.57	0.1309	
D^2^	0.0536	1	0.0536	6.50	0.0232	*
Residual	0.1155	14	0.0083			
Lack of fit	0.1153	10	0.0115	239.23	<0.0001	**
Pure Error	0.0002	4	0.0000			
Cor Total	25.79	28				

* *p <* 0.05; ** *p <* 0.01.

**Table 7 ijerph-19-16848-t007:** Analysis of variance in regression model based on serpentine AR of Cd^2+^.

Source of the Variance	Square-Sum	Degree of Freedom	Average Variance	F Value	*p* Value	Significancy
Model	854.14	14	61.01	15.03	<0.0001	**
A	474.01	1	474.01	116.76	<0.0001	**
B	0.3234	1	0.3234	0.0797	0.7819	
C	103.31	1	103.31	25.45	0.0002	**
D	25.46	1	25.46	6.27	0.0253	*
AB	5.22	1	5.22	1.29	0.2758	
AC	46.04	1	46.04	11.34	0.0046	**
AD	9.18	1	9.18	2.26	0.1549	
BC	0.1640	1	0.1640	0.0404	0.8436	
BD	2.36	1	2.36	0.5804	0.4588	
CD	0.1640	1	0.1640	0.0404	0.8436	
A^2^	150.36	1	150.36	37.04	<0.0001	**
B^2^	0.0062	1	0.0062	0.0015	0.9695	
C^2^	12.32	1	12.32	3.04	0.1034	
D^2^	55.01	1	55.01	13.55	0.0025	**
Residual	56.84	14	4.06			
Lack of fit	56.53	10	5.65	73.41	0.0004	**
Pure Error	0.3080	4	0.0770			
Cor Total	910.98	28				

* *p <* 0.05; ** *p <* 0.01.

**Table 8 ijerph-19-16848-t008:** Verification of experimental results.

Experimental No.	AQ (mg·g^−1^)	AR (%)
1	3.95	95.36
2	3.85	94.11
3	3.93	94.57
Average	3.91	94.68

## Data Availability

Data sharing not applicable.

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
