# Peer review of "Screening and Optimization of Conditions for the Adsorption of Cd^2+^ in Serpentine by Using Response Surface Methodology"

_ijerph, 2022, doi:10.3390/ijerph192416848_

Round 1

Reviewer 1 Report

The authors have used an adsorbent by name serpentine for removal of Cd+. The authors should provide the adsorbent characterization. Otherwise, it is not useful to compare it with other adsorbents. The authors used RSM to optimize the adsorption process.  The concept of the use of DOE for adsorption is not correct. There is nothing called optimal. The authors have to generate the adsorption isotherm covering temperature as parameter and the experiments need to be conducted until the equilibrium is achieved.  The only optimal is the Ph, which need to be experimentally found before generating the adsorption isotherms. The experimental ranges chosen were not significant. The work doesnt add any value to the science or engineering. 

Reviewer 2 Report

The manuscript deals with  optimal conditions for the adsorption of Cd2+ in serpentine from Xiuyan County. It is interesting for the potential application of serpentine in environmental cleaning of heavy metals. Detailed results are presents. However, the specific unsatisfactory points are as follows: 

(1) The adsorption mechanism of the silicate should be discussed.

(2) Some basic characterizations of the used nonmetallic mineral should be added(SEM, XRD, BET, etc.)

Reviewer 3 Report

In this study, the response surface methodology was conducted to screen and optimize the conditions for cadmium adsorption onto serpentine. Overall, this work will draw some interest from scientists in relevant fields. However, some major issues need to be addressed before it is considered for publication.

1. Why did the authors choose the cadmium as the representative heavy metals instead of other analogues such as nickel, lead, zinc, chromium, etc.? Is that a random choice? Please supplement relative background information in the introduction section to illustrate the necessity of cadmium removal. For example, the source and toxicity of cadmium, and the ubiquity of cadmium in water/wastewater, etc.

2. Adsorbent is the key part of the adsorption research, but the information describing the adsorbent, i.e. serpentine in this study, is insufficient. The photograph, SEM images, and BET analysis of the adsorbent utilized in the current study are recommended to provide.

3. How to detect the cadmium concentration in the solutions? Please supplement relevant information including the instrument you used.

4. There should be space between data and unit. Please carefully reexamine the whole manuscript to revise them. For example (but not limited to), Line 91-98, Line 121, Line 150, Line 158, Line 160, etc.

5. Some format errors in References section should be revised.

Reviewer 4 Report

Review on the manuscript with title: Screening and Optimization of Conditions for Adsorption of Cd2+ in Serpentine by Response Surface Methodology. I suggested a major revision with the following comments:

1. In page 2 lines 48-51, the authors’ descriptions for the adsorption effects of natural serpentine are contradictory. It is suggested to add the limitation of natural serpentine as a heavy metal adsorbent, and the reference (J. Clean. Prod. 346 (2022) 131165) maybe helpful for this section and is suggested to be cited. 

2. In page 2 lines 51-56, the modification methods of materials are various. It is suggested that the authors supplement the modification method of natural serpentine and further explain the advantages of thermal activation compared with other methods. The reference (Appl. Catal. B: Environ. 316 (2022) 121639) maybe helpful for this section and is suggested to be cited. 

3. In section 3.1.1 lines 123-125, “resulting in AQ” expressed by the authors in the end is not clear, please explain the reason of AQ increase in more details.

4. In section 3.1.2, the influence of solution pH on the adsorption of materials should consider the changes of surface charge. It is suggested to measure the zero-point potential of natural serpentine and analyze the results, and the reference (J. Hazard. Mater. 423 (2022) 127043) maybe helpful for this section and is suggested to be cited. 

5. In section 3.1.4 lines 187-189, the reason for the rapid increase of adsorption in the first 0.5 hours has not been properly stated. It is suggested to provide a better English expression, and the reference (J. Hazard. Mater. 425 (2022) 127876) maybe helpful for this section and is suggested to be cited. Why was 24 hours chosen when discussing the influence of time on adsorption effect? Why not continue to extend the time?

6. In this study, the thermal activation of natural serpentine is emphasis, which can improve the specific surface area of serpentine and reorganize its internal structure. It is suggested that the author provide some characterization to illustrate the structural changes of material before and after modification, and the reference (J. Hazard. Mater. 432 (2022) 128758) maybe helpful for this section and is suggested to be cited. 

7. The authors mentioned several times in the manuscript that the background of the experiment was industrial wastewater containing cadmium. It is suggested that the authors add a coexisting ion test to explore the potential of the material for practical application, and the reference (Appl. Catal. B: Environ. 310 (2022) 121359) maybe helpful for this section and is suggested to be cited.

8. Why did the author not consider temperature as a factor of the univariate analysis?

Round 2

Reviewer 2 Report

After the revision, the manuscript can be accepted as it is 

Reviewer 3 Report

Good revision.

Reviewer 4 Report

The authors have answered my questions and responded to my comments, and the manuscript has been improved. I recommend the publication of the manuscript in its present form.